# Modeling Local Bond Stress–Slip Relationships of Reinforcing Bars Embedded in Concrete with Different Strengths

**DOI:** 10.3390/ma13173701

**Published:** 2020-08-21

**Authors:** Chao-Wei Tang, Chiu-Kuei Cheng

**Affiliations:** 1Department of Civil Engineering and Geomatics, Cheng Shiu University, No. 840, Chengching Rd., Niaosong District, Kaohsiung 83347, Taiwan; 2Center for Environmental Toxin and Emerging-Contaminant Research, Cheng Shiu University, No. 840, Chengcing Rd., Niaosong District, Kaohsiung 83347, Taiwan; 3Super Micro Mass Research and Technology Center, Cheng Shiu University, No. 840, Chengching Rd., Niaosong District, Kaohsiung 83347, Taiwan; 4Department of Agribusiness Management, National Pingtung University of Science and Technology, No. 1, Shuefu Rd., Neipu, Pingtung 91201, Taiwan; sindy@mail.npust.edu.tw

**Keywords:** local bond stress–slip, bond strength, pullout test

## Abstract

Although many different analytical models of local bond stress–slip have been proposed, considering the possible differences between materials in different countries, their applicability needs to be further explored. In this paper, the local bond stress–slip characteristics of reinforcing bars embedded in concrete with different strengths were experimentally studied. The experimental variables included the concrete strength (20, 40, and 60 MPa) and deformed rebar size (#4, #6, and #8). The experimental results of the bond stress–slip relationship were compared with the Euro-International Concrete Committee (CEB-Comité Euro-International du Béton)-International Federation for Prestressing (FIP-Fédération Internationale de la Précontrainte) Model Code and prediction models found in the literature. In addition, based on the test results, an empirical model of the bond stress–slip relationship was proposed. The evaluation and comparison results show that, regardless of the concrete strength grades, the predicted value calculated using the CEB-FIP Model Code will underestimate the bond strength of the specimens with different steel bar diameters. From this perspective, it is more conservative. In contrast, the proposed model can predict the test results with a reasonable accuracy.

## 1. Introduction

It is well known that reinforced concrete (RC) has many advantages. Therefore, it has become the most important composite material in modern civil engineering and has been widely used in various construction projects. In RC structures, the reinforcing bars are embedded in the tensile area of the concrete. This is mainly to compensate for the tensile strength of the concrete and effectively absorbs all of the tensile forces without separating from the concrete. For an RC member to perform its intended function, it is necessary to generate a bond force at the interface between the concrete and steel to prevent significant slip at the interface. In other words, bond action is the mechanism that ensures the composite behavior of concrete and reinforcing bars [1].

According to the definition of Building Code Requirements for Structural Concrete (ACI 318-11) and Commentary [2], bond stress is a shear stress transmitted along the interface between steel and concrete. For a deformed bar, the capacity of this interface to transfer stresses between the two materials is mainly composed of three resistance mechanisms, namely (1) the chemical adhesion between steel and concrete, (2) the friction between the two surfaces, and (3) the mechanical interlock of the ribs against the concrete, as shown in Figure 1 [3,4]. Once the tensile capacity of the adhesion is exhausted, the embedded bars will slip, resulting in friction. As the slip increases, the friction between steel and concrete diminishes, and the bond strength is sustained through mechanical interaction.

In RC structures, the bond-slip effect between the reinforcing bars and the concrete has a significant effect on the mechanical and seismic performance of the structure and members [5]. On the one hand, the quality of the bond performance directly affects the structure’s deflection, ultimate bearing capacity, crack width, and distribution [3]. On the other hand, under cyclic loading (such as earthquakes), the bond stiffness deteriorates rapidly, resulting in a significant increase in the amount of steel slip. Therefore, the hysteretic performance of RC structures largely depends on the interaction between reinforcing bars and concrete.

Essentially, the magnitude of the bond stress varies along the embedded length of the reinforcing bar. In structural design, several important parameters depend on the bond performance. For example, the anchorage length of the steel bar, the lap splices, and the tension stiffness between the cracks [6]. Moreover, cracks can cause large changes in the bond stress. When a reinforcing bar is stressed in tension, such as in a crack, the resulting elongation can cause sliding relative to the surrounding concrete. The ribs on the periphery of the bar bear on the concrete as they attempt to displace the concrete in order to pullout, thus exerting rupture pressures on the inner perimeter of the concrete [1]. As a result, the crack width and deflection of RC members are affected by the bond stress distribution along the reinforcing bar and the slip between the bar and the surrounding concrete [7].

The bond stress between steel and concrete is a result of the mechanical interaction of the two materials. In structural design, in order to simplify the calculation process, some scholars use the concept of the average bond stress to reflect the steel bond strength with concrete. Eligehausen et al. [8] set a shorter anchor length ($l_{e}<5d_{b}$, where $l_{e}$ is the anchor length and $d_{b}$ is the diameter of the reinforcing bar) in their experimental research. Its purpose is to reduce the influence caused by an uneven bond stress distribution along the embedded length of the steel bar. Therefore, that the measured test results could be close to the true local bond stress. Oh and Kim [9] proposed a realistic model of the bond stress-slip relationship under repeated loads. The test showed that if bond failure did not occur, the bond strength and the slip at peak bond stress were not influenced much by repeated loading. However, the values of loaded slip and residual slip increased with increasing load cycles. In a theoretical analysis, Filippou et al. [10] assumed that the bond stress was evenly distributed along the anchoring length, and the following differential relationship could be established according to the balance between reinforced concrete:(1)$$∆F=\frac{\pi d_{b}^{2}}{4}d\sigma_{s}=\tau \pi d_{b}dx$$
where τ is the bond stress and $\sigma_{s}$ is the reinforcing bar stress. Then, the bond stress derivation formula could be obtained:(2)$$\tau \left(x\right)=\frac{∆F}{\pi d_{b}∆x}=\frac{d_{b}}{4}\frac{d\sigma_{s}}{dx}.$$

Basically, bond stress is a function of several parameters, such as concrete’s compressive strength, the surface of the bar roughness and/or irregularities, the diameter of the bars, and the type and disposition of the ribs. As early as 1987, it was proposed that bond stress–slip–rebar strain can be used to express the bond–slip relationship [11]. Kankam [12] established the relationship between bond stress, steel bar stress, and slip based on an experimental analysis. At present, there are two common ways to express the bond–slip relationship. Among them, *τ*-*s* is the most common, that is, the relationship between the bond stress and the amount of slip. The other is *σ*-*s*, which is the steel bar stress and slip. The two can be transformed into each other.

In the experimental research on the bond-slip relationship of reinforced concrete, there are two commonly used test methods: the pull-out test and the beam test [13]. Among them, the pull test is the main research method. Eligehausen et al. [8] conducted a pullout test on 125 beam–column joint specimens through displacement control loading. The test comprehensively considered the influence of various factors on the bond–slip relationship, including the steel bar diameter, concrete strength, transverse compressive stress, load rate, etc. They finally established the bond–slip constitutive relationship of deformed bars under a monotonic load and cyclic load. Filippou et al. [10] established an analytical model describing the hysteretic behavior of reinforced concrete beam-column joints. The model considered the effect of bond slip and the degradation of bond strength under the control of cyclic displacement. Therefore, the model is collectively referred to as the Eligehausen–Filippou model. CEB-FIP Model Code 1990 [14] and 2010 [15] also adopted this model successively, which comprised four distinct branches: A curvilinear ascending region; a constant maximum region; a linearly descending region; and a region of constant frictional bond stress, as shown in Figure 2. In order to establish the bond–slip constitutive relationship of moderately confined concrete, Guizani et al. [16] produced 43 moderately anchored reinforced concrete specimens. The anchorage length of each specimen was $5d_{b}$, and the influence of the degree of restraint, the layering effect of concrete, the amount of initial loading, etc., on the test was studied. Based on Eligehausen’s test, Xu [17] conducted a pullout test on 334 reinforced concrete specimens with variables such as the concrete strength, concrete cover thickness, and stirrup content. Finally, using the experimental data for regression analysis, the calculation formula of the characteristic strength and the corresponding bond–slip constitutive relationship were obtained.

In view of the importance of the bond between steel and concrete, many scholars have conducted extensive research. In the past few decades, many studies have proposed characteristic parameters for this complex problem [18,19,20,21,22,23,24,25,26,27,28,29,30,31,32,33,34,35,36,37,38,39,40,41,42,43,44,45,46,47,48]. However, due to the complexity of the bond interface, data dispersion, and other test conditions, the bond–slip constitutive relationship proposed by different scholars has certain differences and is often only applicable to specific situations. Therefore, further research and the application of such formulas are restricted. Based on the above analysis, it is necessary to conduct an in-depth study of the mechanism and factors influencing bond slip, as well as establish the corresponding bond–slip relationship. Its purpose is to reveal the behavior response of the RC structure more accurately and further improve the performance of the RC structure. Under the conditions of different rebar sizes and concrete strengths, this study performed the pullout experiment to compare and analyze the effect of each factor. In addition, through regression analysis, the formula for the local ultimate bond stress between the deformed steel bar and the concrete was obtained. Furthermore, a modified local bond stress–slip relationship was proposed.

## 2. Experimental Program

### 2.1. Material Properties

The materials used in this study for making specimens included cement, fine and coarse aggregates, superplasticizer, and reinforcing steel. The cement used was locally produced Type I Portland cement with a specific gravity of 3.15 and a fineness of 3400 cm^2^/g. Aggregate was locally produced. The coarse aggregate was crushed stone with a maximum particle size of 19 mm. the fine aggregate was natural river sand. Their physical properties are listed in Table 1. Two different superplasticizers produced locally (HICON HPC 1000 for medium-strength concrete and HICON MTP A40 for high-strength concrete) were used. HPC 1000 complies with the American ASTM C494 Type D regulations and HICON MTP A40 complies with the American ASTM C494 Type G regulations. Their physical properties are shown in Table 2. Three different reinforcing bars (#4, #6, and #8) of A706 were used. Their physical and mechanical properties are shown in Table 3.

### 2.2. Concrete Mix Design and Test Specimens

The literature shows that the bond strength is closely related to the compressive strength of concrete. To analyze the influence of the concrete strength on the bond behavior, the specified 28-day compressive strengths chosen were equal to 20, 40, and 60 MPa (representing low, medium, and high grades respectively). This study referred to the Standard Practice for Selecting Proportions for Normal, Heavyweight, and Mass Concrete [49] specification for concrete mix design and adjusted the composition after trial mixing. The concrete mix design used is shown in Table 4. The abbreviations for identifying each concrete indicate the type of concrete: normal-aggregate concrete (C) and the strength of concrete (30 or 50 MPa).

All aggregates were cured indoors until the required saturated surface-dry condition was reached. The processed aggregates were then stored indoors, and the ambient temperature and relative humidity (RH) were controlled at 25 °C ± 3 °C and 50% ± 5%, respectively, to prevent moisture changes in the aggregate. During mixing, the cement, fine aggregates, and coarse aggregates were first blended in the biaxial mixer at a rate of 45 revolutions per minute for about 1 min. Then the water and superplasticizer were added in the mixer and blended for about 1.5 min.

The pullout specimens were 150 mm cubes, which were cast using steel molds. For the cubic specimen, its single bar was anchored vertically along the central axis (see Figure 3). The embedded length in the pullout specimens was three times the bar diameter (i.e., $l_{e}=3d_{b}$). As suggested by Soroushian et al. [36], this embedded length was short enough to assume that the slippage recorded was representative of a local bond stress value. The unbounded regions of the bar were sheathed with PVC pipes. In addition, the specimen contained three transverse stirrups to limit its splitting when the bar was placed in tension. In this study, three kinds of concrete strengths were tested, and three kinds of steel bars with different diameters were respectively configured to make nine groups of pullout specimens. To check the reliability of the test method and the dispersion of the test results, two nominally identical samples were made in each group.

When casting the pullout specimen, the fresh concrete was slowly poured into the mold to a depth of half, which was followed by controlled vibrations. After fully vibrating the sample, the mold was filled with concrete and vibrated again to ensure that the concrete was well-compacted. For each concrete mixture, six cylindrical specimens of 100 mm diameter × 200 mm height were also cast, hereinafter referred to as control cylinders. In addition, for each concrete mixture, six cylindrical specimens with a diameter of 150 mm × 300 mm in height were cast to determine their split tensile strength. After casting, the specimens were covered overnight with wet hessian and polyethylene sheets for a period of 24 h. Then, the pullout specimens and their respective control cylinders were removed from the molds. To maintain the same environmental conditions, all specimens were placed in water containers in the laboratory for 27 days. Each test was conducted 28 days after casting.

### 2.3. Instrumentation and Testing Procedures

According to Standard Test Method for Slump of Hydraulic-Cement Concrete ASTM C143 [50], Standard Test Method for Compressive Strength of Cylindrical Concrete Specimens ASTM C39 [51], Standard Test Method for Splitting Tensile Strength of Cylindrical Concrete Specimens ASTM C496 [52], and Standard Test Method for Static Modulus of Elasticity and Poisson’s Ratio of Concrete in Compression ASTM C469 [53], the slump, compressive strength, splitting strength, and elastic modulus of concrete were tested, respectively. The pullout specimens were loaded by using a 500 kN MTS servo valve-controlled machine, which was equipped with a special test frame, as shown in Figure 4. It can be seen from Figure 4 that one end of the test bar was loaded, and no load was applied at the other end. Three linear variable differential transformers (LVDTs) were used to measure the relative bond slip between steel and concrete. The test setup is shown in Figure 4. The pullout force was applied under displacement control at a constant displacement rate of 0.01 mm/sec until the specimen failed. The pullout force was measured by a dynamometer installed in the testing machine. The test progress was monitored on a computer screen. In addition, all load and displacement data were captured and stored on a floppy disk through a data logger.

In most pullout tests, the embedded length of the steel bar was set to be short ($l_{e}\leq 5d_{b}$), the steel bar was basically maintained in the elastic stage, and the bond stress was approximately constant [5]. In view of this, this study assumed that the bond stress was uniformly distributed along the embedded length (see Figure 5). Therefore, the bond stress could be calculated by dividing the applied load by the contact area between the steel bar and the concrete, as shown in the following equation:(3)$$\tau =\frac{P}{\pi d_{b}l_{e}}$$
where *τ* is the bond stress (MPa), *P* is the applied load (N), $s_{l}$ and $d_{b}\mathrm{represent}\;\mathrm{the}$ bar diameter (mm), and $l_{e}$ is the embedded length (mm).

On the other hand, this value is usually obtained by measuring the steel bars at both ends of the specimen. Therefore, the analysis model of bond slip could be directly derived from the test data. The relative slip of the rebar and concrete corresponding to the bond stress could be divided into the slip at the loading end $\left(s_{l}\right)$ and the slip at the free end $\left(s_{f}\right)$. In the case of a local bond, the relative slip of the rebar and concrete can be regarded as rigid motion, so under the same load, $s_{l}$ and $s_{f}$ should be the same. In this study, the average value of $s_{l}$ and $s_{f}$ could be taken as the slip corresponding to the bond stress, as shown in the following equation:(4)$$s=\frac{s_{l}+s_{f}}{2}.$$

## 3. Experimental Results and Discussion

### 3.1. Fresh and Mechanical Properties of Concrete

The slump values of each concrete mixture are listed in Table 5. It can be seen from Table 5 that the slump of the three groups of concrete proportions is approximately the same, about 16–17 cm. On the day of the pullout test, each control cylinder was capped and subjected to a compression test to determine the compressive strength of each concrete mixture. The compressive strength of each concrete mixture is the average of three specimens. The test results show that the average value of the 28-day compressive strengths is close to the target value (that is, 20, 40, and 60 MPa), as shown in Table 5. In addition, the average values of the splitting tensile strength and elastic modulus of each concrete mixture are also listed in Table 5.

### 3.2. Local Bond Stress–Slip Behavior

In this paper, the bond stress was calculated according to Equation (3). In addition, according to Equation (4), the slip was calculated as the average motion of the loaded end and free end of the steel bar relative to the concrete block. Therefore, the bond–slip curve could be drawn directly based on the test data. The local bond stress–slip curves for specimens with different concrete strengths and rebar diameters are presented in Figure 6, Figure 7 and Figure 8. With the different test variables, the bond stress–slip relationship of each specimen was also different. Overall, the behavior of the bond stress–slip relationship of the specimen shows that when the bond stress initially increased, the slip was very small. However, once the maximum bonding stress was reached, the curve softened immediately. During the loading process of the pullout test, the load was mainly transferred between the reinforcement and the surrounding concrete by means of adhesion and a mechanical bond. However, due to the low tensile strength of the interfacial zone, the bond strength, which depends on adhesion and surface friction, was inherently weak. In contrast, the inclined ribs acted as an interlocking bearing on adjacent concrete, and the deformed rebar produced a greater bond strength. This is the main mechanism of load transfer, and its strength limit state is usually controlled by the splitting strength of surrounding concrete [19]. During the translation of the deformed bar relative to the concrete, the ribs will either split the concrete by pushing it away or crush the concrete by enclosing it in the spaces between them. Lutz et al. [4] have shown that, for rib angles between 40° and 105°, relative motion is almost entirely caused by the latter effect.

From the pullout test results, it can be seen that the response can be divided into five stages (see Figure 9), as described below:
Stage 1: At the initial stage of loading, there was a short non-slip straight line section, mainly due to the chemical adhesion and friction between the steel bar and the concrete;Stage 2: When the loading was up to $\tau_{1}$/$\tau_{u}$ ≈ 0.3, the chemical adhesion between the rebar and concrete failed, and the rebar and concrete started to produce a relative slip;Stage 3: When the loading increased continually and the bond stress reached a splitting bond stress ($\tau_{cr}$), radial splits appeared around the rebar due to the radial pressure exerted by the rebar lugs. However, the confining effect of the stirrup could delay the development of splitting, and the load could continue to increase;Stage 4: When the loading was increased continually for the ultimate bond stress ($\tau_{u}$), the concrete within the clear rib spacing of the tested rebar was crushed completely, and shear-cut slip along the rib outer diameter occurred. At this time, the bond stress decreased quickly and the slip amount increased quickly;Stage 5: When the slip amount reached $s_{3}$, approximately the clear rib spacing of the tested rebar, the concrete between the ribs was completely cut-off. At this time, the bond stress did not keep decreasing; that is, the residual bonding stress ($\tau_{f}$) was only provided by the friction between the steel bar and the surrounding concrete.

### 3.3. Failure Modes

Bond failures are generally divided into pullout failures or splitting failures. In this study, the failure of most test specimens belonged to the pull-out mode, while a few specimens failed in splitting failure mode. Due to adhesion, the bond stress–slip curve was initially very steep, as shown in Figure 6, Figure 7 and Figure 8. It is worth noting that the magnitude of the bond stress did change according to the test variables, and the effect of the size of the rebar is the most significant. Regardless of the concrete strength level, all specimens with the #4 rebar or #6 rebar exhibited pullout failure, which resulted in shearing along a surface at the top of the ribs around the rebar. The reason for this failure mode is that the stirrup was able to prevent or delay splitting failure. In other words, on the one hand, when the radial force from the loaded rebar is less than the resistance of the surrounding concrete and/or stirrups, and on the other hand, the tangential force is greater than the resistance of the concrete, a pullout failure mode will occur.

For the specimen with a concrete strength of 60 MPa and a #8 rebar, the concrete cover was less than that of the #4 rebar, and under the condition of an embedded length equivalent to $3d_{b}$, the embedded length of the rebar was longer than that of the #4 rebar specimen. As the rebar was loaded, it exerted radial pressure on the surrounding concrete. When the bond stress developed to a fixed level, if the surrounding concrete or stirrups were not able to resist the pressure, cracks were generated at the interface of the concrete-reinforcing steel. Once these cracks propagated to the surface, they caused failure of the concrete by concrete cover splitting, as shown in Figure 10. This caused failure as a result of split before developing into a real ultimate bond stress. As can be seen from Figure 8, the bond stress decreased dramatically when the specimens produced splitting failure. This meant that the rebar and concrete lost their bond performance. As a result, only the frictional force between the rebar and the surrounding concrete could provide residual bond stress.

### 3.4. Ultimate Bond Stress

Pull out tests were conducted on specimens with different concrete strengths and rebar diameters. In total, there were nine different combinations of concrete strength and rebar diameter. A total of 18 specimens completed the pullout test, and the test age was 28 days. The bond strength is a key mechanical property of RC members, which refers to the bond stress corresponding to the ultimate load recorded during the test. Table 6 presents the experimental results of ultimate bond stress $\tau_{u}$ for each combination of concrete strength and rebar diameter, which is the average of two nominally identical test specimens.

#### 3.4.1. Comparison of Ultimate Bond Stress between Test Result and Prediction Models

Investigations into the bond strength between the concrete and rebar have been carried out for many years. Many researchers have studied the relationship between pullout load and compressive strength. All studies in this field unanimously show that the compressive strength of concrete is an important factor affecting the bond strength [15,37]. In addition, researchers have also explored the influence of other factors, such as the thickness of the concrete cover, the diameter of the rebar, the embedded length, and the transverse stirrups ratio, on the bond strength [38,39,40]. This study selected some common prediction models of bond strength to compare with the test results. Further, this study proposes two prediction models of bond strength based on the test results. These prediction modes are described below.

(1) CEB-FIP Model Code 2010

For specimens with good confinement and non-split failure, the calculation formula of the ultimate bond stress (bond strength) $\tau_{u}$ recommended by the CEB-FIP Model Code 2010 is as follows [15]:(5)$$\tau_{u}=2.5\sqrt{f_{c}^{\prime }}\;(\mathrm{MPa}),$$
where $f_{c}^{\prime }$ is the concrete compressive strength.

(2) The prediction model proposed by Huang et al.

Huang et al. proposed a prediction formula for the ultimate bond stress, as shown below [37]:(6)$$\tau_{u}=0.45f_{c}^{\prime }\;(\mathrm{MPa})$$
where $f_{c}^{\prime }$ is the concrete compressive strength.

(3) The prediction model proposed by Xu

Xu proposed a prediction formula for the ultimate bond stress, as follows [17]:(7)$$\tau_{u}=\left(0.82+0.9\frac{d_{b}}{l_{e}}\right)\left(1.6+\frac{0.7c}{d_{b}}+20\rho_{sv}\right)f_{t}\;(\mathrm{MPa}),$$
(8)$$f_{t}=0.267f_{c}^{\prime 2/3}\;\left(\mathrm{MPa}\right),$$
(9)$$\rho_{sv}=\frac{A_{sw}}{CS_{w}}$$
where $f_{c}{}^{\prime }\;\mathrm{is}\;\mathrm{the}$ compressive strength of concrete, *c* is the thickness of concrete cover, $d_{b}\;\mathrm{is}\;\mathrm{the}$ diameter of the rebar, $l_{e}$ is the embedded length, $f_{t}\;\mathrm{is}\;\mathrm{the}$ tensile strength of concrete, $\rho_{sv}$ is the stirrup ratio, $A_{sw}$ is the stirrup area, and $S_{w}$ is the stirrup spacing.

(4) The prediction model proposed by Soroushian and Choi

Soroushian and Choi proposed the empirical formula of the ultimate bond stress from the partial bond pull test, as shown below [38]:(10)$$\tau_{u}=\left(20-d_{b}/4\right)\sqrt{f_{c}^{\prime }/30}\;(\mathrm{MPa}),$$
where $f_{c}{}^{\prime }\;\mathrm{is}\;\mathrm{the}\;$ compressive strength of concrete and $d_{b}$ is the diameter of the rebar.

(5) The prediction model proposed by Aslani and Samali

Aslani and Samali proposed an empirical formula for the ultimate bond stress of concrete, as follows [39]:(11)$$\tau_{u}=\left[0.679{\left(\frac{c}{d_{b}}\right)}^{0.6}+3.88\left(\frac{d_{b}}{l_{e}}\right)\right]{\left(f_{c}^{\prime }\right)}^{0.55}\;(\mathrm{MPa}),$$
where $f_{c}{}^{\prime }$ is the compressive strength of concrete, *c* is the thickness of concrete cover, $d_{b}$ is the diameter of the rebar, and $l_{e}$ is the embedded length.

(6) The prediction model proposed by this study

According to the test results, this study proposes two prediction models. One of them is obtained by simple regression analysis and the other is obtained by multiple regression analysis, which are described as follows:(12)$$\tau_{u}=8.9824e^{0.0193f_{c}^{\prime }}\;(\mathrm{MPa}),$$
(13)$$\tau_{u}=0.384702f_{c}^{\prime }-1.73018d_{b}-7.40325\frac{c}{d_{b}}+65.90284\;(\mathrm{MPa})$$
where $f_{c}{}^{\prime }\;\mathrm{is}\;\mathrm{the}\;$ compressive strength of concrete, *c* is the thickness of concrete cover, and $d_{b}$ is the diameter of the rebar. It can be seen from Figure 11 that the proposed univariate regression prediction model of bond strength has a good correlation coefficient. The coefficient of determination (*R*^2^) of Equation (13) is 0.9568.

The experimental results of $\tau_{u}$ for each specimen, together with their values predicted by each prediction model, are compiled in Table 7. It can be seen from Table 7 that regardless of concrete strength grades or the rebar size, the measured $\tau_{u}$ is greater than the $\tau_{u}$ recommended by the CEB/FIP Model Code. In other words, the CEB/FIP Model Code is more conservative. The reason for this result can be seen from Equation (5); the CEB/FIP Model Code only considers the effect of the concrete compressive strength and does not consider other factors. Huang et al.’s prediction model is also more conservative. Therefore, the CEB/FIP and Huang et al.’s prediction model [37], which only consider the effect of the concrete strength on $\tau_{u}$, are unable to accurately estimate the test results. In addition to considering the compressive strength of concrete, the prediction models of Xu [17], Soroushian and Choi [38], and Aslani and Samali [39] also include the influence of parameters such as concrete cover, the rebar diameter, and embeddedness. The prediction results of these prediction models tend to be overestimated, but there are also cases where underestimation occurs, as shown in Table 7. In contrast, the two prediction models proposed in this study have excellent prediction results, as shown in Figure 12. It can be seen from Figure 12 that the less scatter of data around the diagonal line confirms the fact that Equations (12) and (13) are an excellent predictor for the value of $\tau_{u}$. The correlation between the experimental values and the predictive values, which were obtained from Equations (5) and (7) and Equations (10) and (11), is more scattered.

#### 3.4.2. Effect of the Rebar Size on the Ultimate Bond Stress

In theory, the rib area increases with an increasing rebar diameter. However, the shape parameter of the rebar does not change linearly with the rebar diameter, and the height of the rib decreases relatively. Accordingly, the area of the rib does not actually increase much. In other words, the larger the diameter of the rebar, the smaller the bond area. As a result, $\tau_{u}$ is smaller. This result can be confirmed by Figure 13, regardless of the concrete strength of the test specimen. When the diameter of the rebar changed from #6 to #8, the normalized ultimate bond strength ($\tau_{un}=\tau_{u}/\sqrt{f_{c}^{\prime }}$) decreased.

In fact, the rib height also plays a vital role when discussing the bond properties between the concrete and rebar. Figure 13 shows that under the same concrete strength conditions, the $\tau_{u}$ of specimens with #4 and #8 were lower than that of specimens with #6. The reason for this is that the $\tau_{u}$ of the specimen is significantly related to the rib height of the rebar. As can be seen from Table 3, the rib height values of #4, #6, and #8 are 0.7, 1.9, and 1.7 mm, respectively. Furthermore, the ratio of the rib height to diameter ($h/d_{b}$) of the rebar can be used as a parameter to explore its effect on $\tau_{u}$. According to the physical properties of steel used in this study (see Table 3), the values of $h/d_{b}$ for #4, #6, and #8 are 0.055, 0.1, and 0.067, respectively. It can be seen from Figure 14 that, for specimens that failed in the pullout mode, the greater the value of $h/d_{b}$, the greater the $\tau_{u}$. From this perspective, the use of $h/d_{b}$ in analyzing $\tau_{u}$ not only demonstrates a regular linear relationship, but also interprets test results well.

#### 3.4.3. Effect of the Concrete Strength on the Ultimate Bond Stress

Table 5 shows that as the compressive strength of concrete increases, its splitting strength also increases. Therefore, along with the increase in compressive strength, the bond performance between the rebar and the concrete also increases, developing higher bond stress before failure, as shown in Figure 15. Under the condition of a lower concrete strength, the aggregate in the cracked surface is mostly not damaged, which is able to provide a higher bearing characteristic, thus enhancing the overall bond strength. However, under the condition of high-strength concrete, the aggregate in the cracked surface is often cleaved or broken. The bearing force provided by aggregates is relatively small, so the mortar matrix is the main factor affecting bond failure.

### 3.5. Comparison of Measured Bond Stress–Slip Behavior with Prediction Models

As early as 1983, Eligehausen et al. [8] developed a bond–slip analysis model based on a single pull test with a short embedding length. It is arguably one of the most widely used and accepted bond–slip models. The CEBFIP Model Code 2010 also uses this model. The analytical bond stress–slip model recommended by the CEB-FIP Model, Huang et al. [37], and Harajli et al. [40] is shown in Figure 16. It is composed of four distinct branches. The initial ascending branch reaches the ultimate bond stress ($\tau_{u}$) for $s\leq s_{1}$. The second branch is regarded as a plateau, during which the slip continues to increase while maintaining a constant bonding stress. $\left(\tau =\tau_{u}\right)$ for $s_{1}<s\leq s_{2}$. Then, the linear descending branch begins from $\left(s_{2},\;\tau_{u}\right)$ to $\left(s_{3},\;\tau_{f}\right)$. Subsequently, a constant residual bond strength ($\tau_{f}$) is reached due to pure friction between the rebar with cracked concrete lugs and surrounding concrete. In this study, these three modes were selected for comparative analysis with the measured bond stress–slip behavior of the specimen. According to the bond stress–slip relationship recommended by CEB-FIP Model, Huang et al. [37], and Harajli et al. [40], the bond stress (*τ*) between the concrete and rebar can be calculated as a function of the relative displacement (*s*) using the following equation:(14)$$\tau \left(s\right)=\{\begin{array}{c} \tau_{u}{\left(s/s_{1}\right)}^{\alpha }\;\mathrm{for}\;0\leq s\leq s_{1}\\ \tau =\tau_{u}\;\mathrm{for}\;s_{1}<s\leq s_{2}\\ \tau =\tau_{u}-\left(\tau_{u}-\tau_{f}\right)\left(\frac{s-s_{2}}{s_{3}-s_{2}}\right)\;for\;s_{2}<s\leq s_{3}\\ \tau =\tau_{f}\;\mathrm{for}\;s_{3}<s\; \end{array}$$
where $\tau_{u}$ is the ultimate bond stress, $\tau_{f}$ is the residual bond stress; *s* is the bond slip, $s_{1}$, $s_{2}$, and $s_{3}$ are the slip at the beginning of ultimate bond stress, slip at the end of ultimate bond stress, and slip at the beginning of residual bond stress, respectively, and *α* is a curve fitting parameter, and its value is less than 1.

The controlling parameters suggested by CEB-FIP, Huang et al. [37], and Harajli et al. [40] are shown in Table 8. According to Table 8, although the controlling parameters are different, the main curves of these three bond stress slip models are basically similar to each other. On the other hand, based on the test results, this study replaces $\tau_{u}$ in Table 8 with Equation (13). In addition, the relationship between the observed slip corresponding to the maximum bond strength $\left(s_{1}\right)$, the clear rib spacing of the tested rebar $\left(s_{3}\right)$, and the rebar diameter $\left(d_{b}\right)$ is obtained by regression analysis, as shown in Equation (15). Then, other controlling parameters are also adjusted according to the actual results. Through this amendment, a bond–slip model is proposed, and its controlling parameters are shown in Table 8.
(15)$$s_{1}=s_{3}\left(0.025685s_{3}-0.04244d_{b}+0.566648\right)\;(\mathrm{mm})$$

The bond stress–slip relationship recommended by CEB-FIP Model, Huang et al. [37], and Harajli et al. [40] was compared with the measured bond stress–slip behavior of the specimen (see Figure 17, Figure 18 and Figure 19). As can be seen from these figures, regardless of the strength of the concrete, the predicted value calculated using the CEB-FIP Model Code and Harajli et al.’s model underestimates the bond strength of the specimen with different rebar diameters. From this perspective, these two models are more conservative. In addition, it can be seen that the predicted value of low-strength concrete calculated by the model of Huang et al. underestimated the bond strength in all cases, while the predicted value of medium-high-strength concrete was overestimated. Theoretically, residual bond stress is associated with the ultimate bond. Therefore, when there is some difference between the ultimate bond stress and the predicted values calculated with the three models, the residual bond stress will also exhibit some differences. As for the value of slip at the start of residual bond stress, the measured slip is close to the value suggested by the three models, which is equal to one bar rib-net spacing. On the other hand, the slope of the curve for all specimens is far greater than those specified in the three models. This means that under the same bond stress, the measured slip is smaller than the value specified in the three models.

As for the prediction model proposed by this research, although the prediction results of medium strength concrete are overestimated, the difference is limited; while the prediction results of low- and high-strength concrete are underestimated, the difference is quite limited. Overall, the proposed prediction model is more accurate and has reference value.

### 3.6. Local Slip

In the local bond stress–slip model suggested by the CEB/FIP Model Code, the bond stress–slip curve is initially very steep. In particular, the bond stress–slip relationship is quite non-linear near the ultimate bond stress. In fact, before the stress reaches the ultimate value, cracks in the concrete will also affect the slip. In other words, it is difficult to obtain good and accurate results by only comparing slip $s_{1}$. In view of this, this study assumes that when the sliding value is between zero and $0.4s_{1}$, the bond stress–slip relationship is linear, as shown in the following equation:(16)τ(s)=κ·s(x),
where τ(s) is the local bond stress (only a function of the relative slip), s(x) is the relative slip, and *κ* is the bond modulus (the paper takes $\kappa =\tau /0.4s_{1}$). In essence, the bond modulus has a physical meaning. If the bond stress is maintained at a certain value, the larger the value of the bond modulus, the smaller the slip. Therefore, under the condition that each variable is fixed, by applying *κ* to compare the difference between the bond stress–slip relationship, the result will be clearer.

The effect of concrete’s compressive strength on the bond modulus is shown in Table 9. It can be clearly seen from Table 9 that the bond modulus increased with the strength of concrete. The reason for this is that due to the increase in concrete strength, the bond properties between steel and concrete became better, so the relative sliding between concrete and steel was smaller. As can be seen from Figure 20, under the condition of a fixed concrete strength, the bond modulus of the specimen with #6 was the largest, the bond modulus of the specimen with #8 was the smallest, and the bond modulus of the specimen with #4 was between the two. From this point of view, in order to ensure objective results, the percentage of the rib area to the overall bond area should be considered.

It can also be seen from Figure 20 that when the steel bar size changes from #6 to #8, the bond modulus attenuation of the specimen with high strength concrete is much larger than that of the specimen with low and medium strength concretes. This result clearly shows that under the same bonding stress, the slip value of the specimen with #8 will be much higher than that of the specimen with #6. In other words, the bond–slip behavior of high-strength RC members in the presence of tensile cracks will change significantly, depending on the size of the steel bar.

This study further analyzed the ascending branch of the local bond stress–slip curve of each specimen and derived the equation of the fitting curve. Through regression analysis, the local bond stress–slip curve before the ultimate bond strength was obtained. The coefficients of determination of the regression equation were mostly above 0.90, as shown in Figure 21 (taking specimens with the #8 rebar as an example). By establishing the relationship between the local bond stress–slip relationship and the bond stress–slip relationship of the steel bar along the embedded length, the true bond stress–slip relationship curve at the anchor position of the steel bar could be obtained, that is, the position function ϕ(x). On the other hand, the establishment of the local bond stress–slip relationship has a considerable degree of benefit for the application of finite element analysis.

## 4. Conclusions

According to the analysis results of the experiment, the following conclusions can be obtained:Regardless of the concrete strength grades, the predicted value calculated using the CEB-FIP Model Code and Harajli et al.’s model underestimates the bond strength of the specimen with different rebar diameters. From this perspective, these two models are more conservative.The predicted value of low-strength concrete calculated by the model of Huang et al. underestimated the bond strength in all cases, while the predicted value of medium-high-strength concrete was overestimated.The established empirical formula for local ultimate bond stress is a function of the concrete strength, thickness of concrete cover, and diameter of the rebar, which fits well with the experimental values.The analysis and comparison of the calculated values and the test results confirm that the proposed model can reasonably predict the local bond behavior for different rebar diameters and concrete strength grades.

## Figures and Tables

**Figure 1 materials-13-03701-f001:**
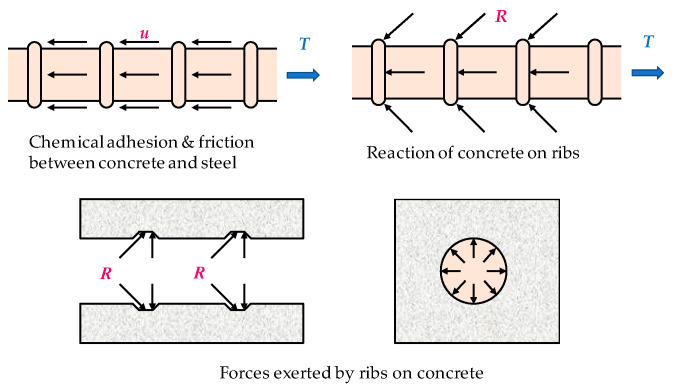
Bond force transfer mechanisms.

**Figure 2 materials-13-03701-f002:**
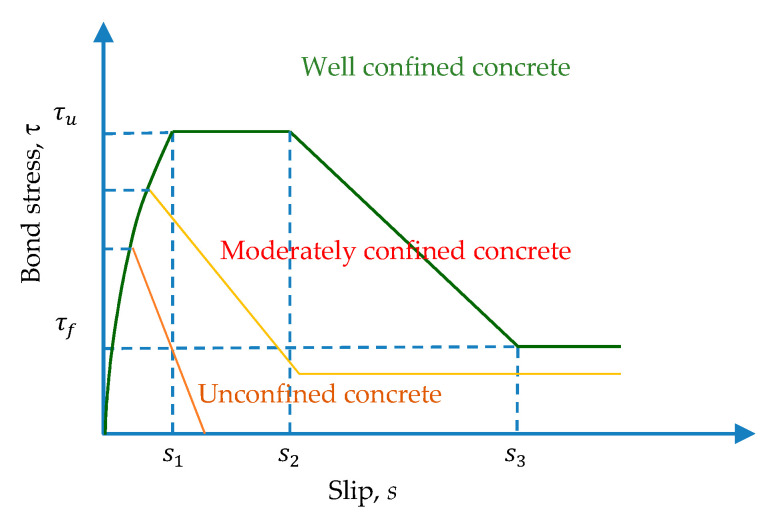
Comparison of bond–slip curves under different constraints.

**Figure 3 materials-13-03701-f003:**
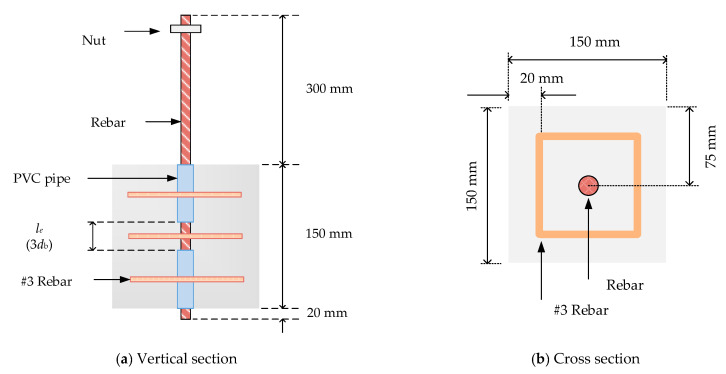
Dimensions and cross-sections of specimen: (**a**) Vertical section; (**b**) Cross section.

**Figure 4 materials-13-03701-f004:**
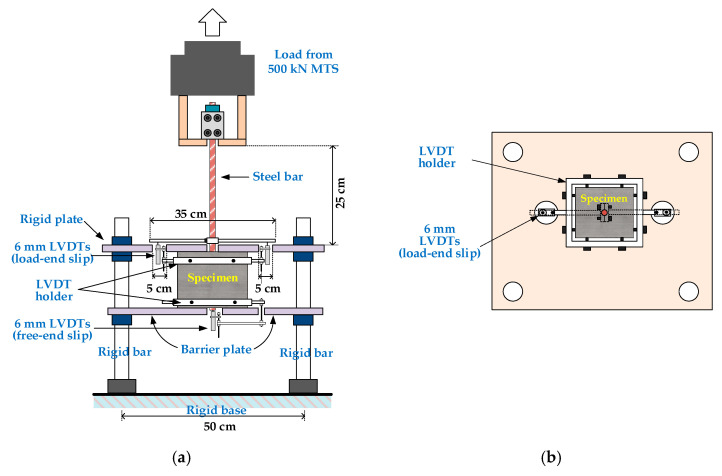
Setup of the pullout test: (**a**) front view; (**b**) top view.

**Figure 5 materials-13-03701-f005:**
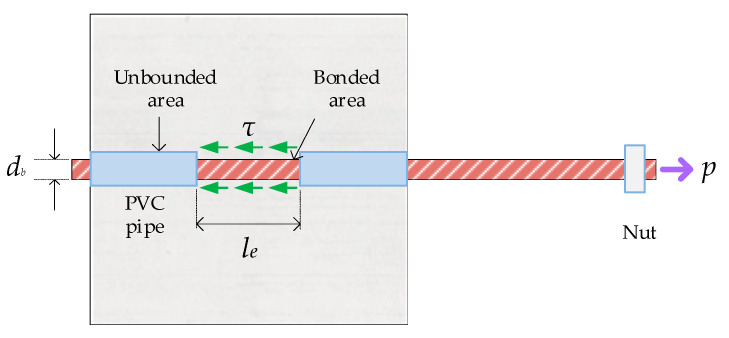
Schematic diagram of local bond stress between the bar and concrete.

**Figure 6 materials-13-03701-f006:**
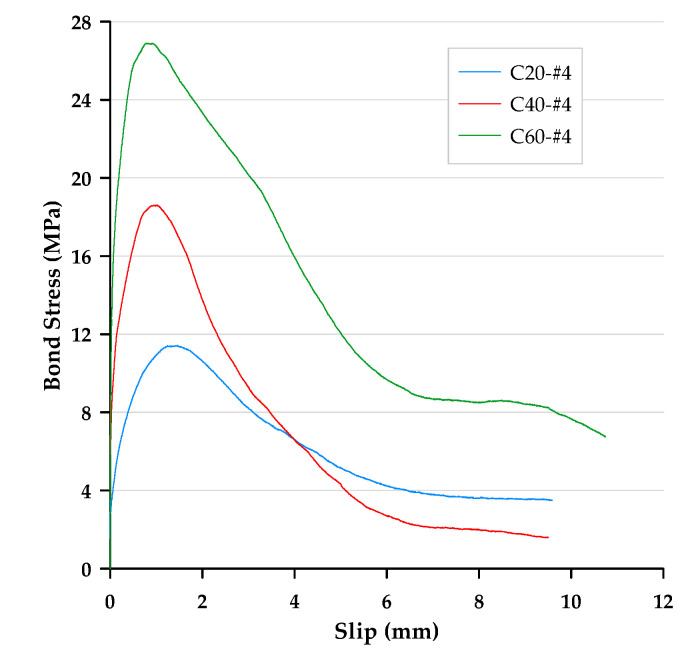
Local bond stress–slip curve for specimens with the #4 rebar.

**Figure 7 materials-13-03701-f007:**
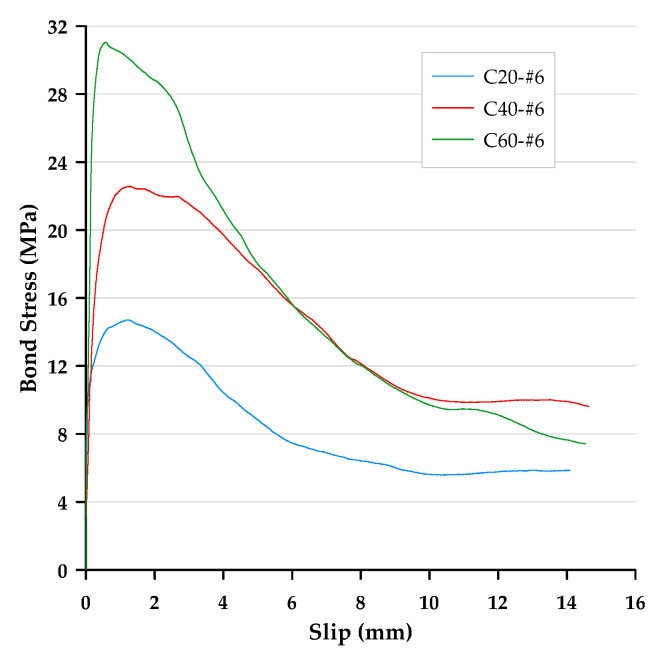
Local bond stress–slip curve for specimens with the #6 rebar.

**Figure 8 materials-13-03701-f008:**
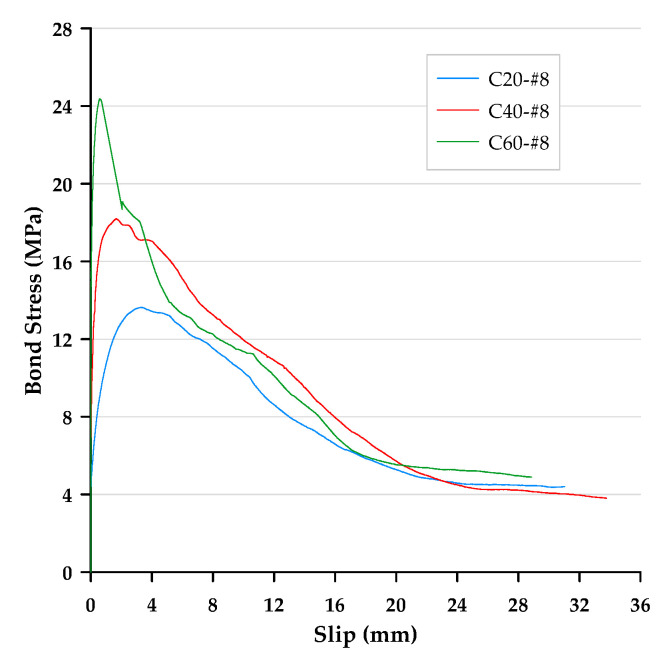
Local bond stress–slip curve for specimens with the #8 rebar.

**Figure 9 materials-13-03701-f009:**
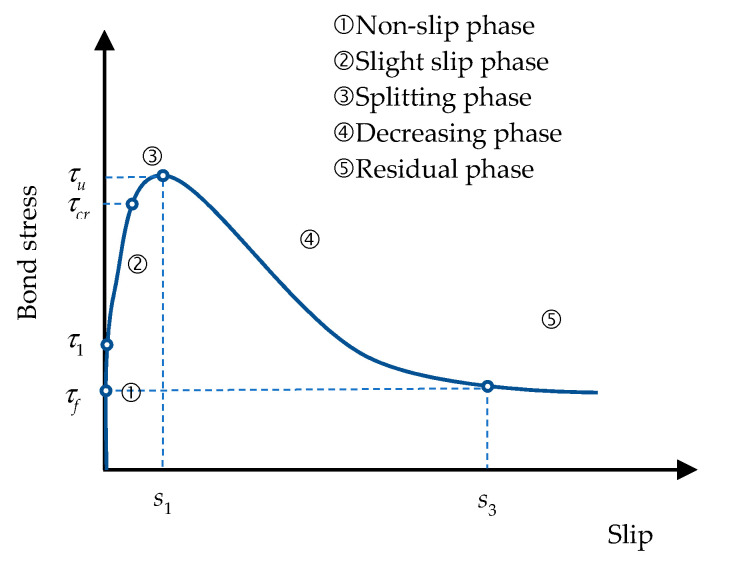
Typical local bond stress–slip curve.

**Figure 10 materials-13-03701-f010:**
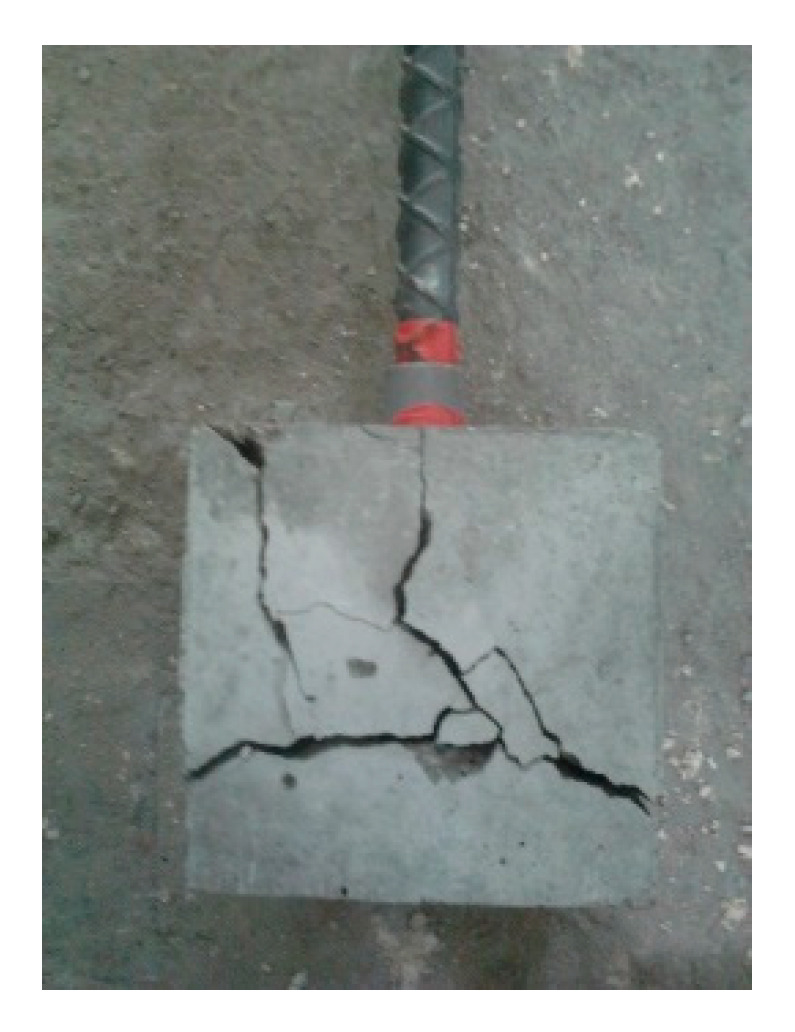
Appearance of the specimen with splitting failure.

**Figure 11 materials-13-03701-f011:**
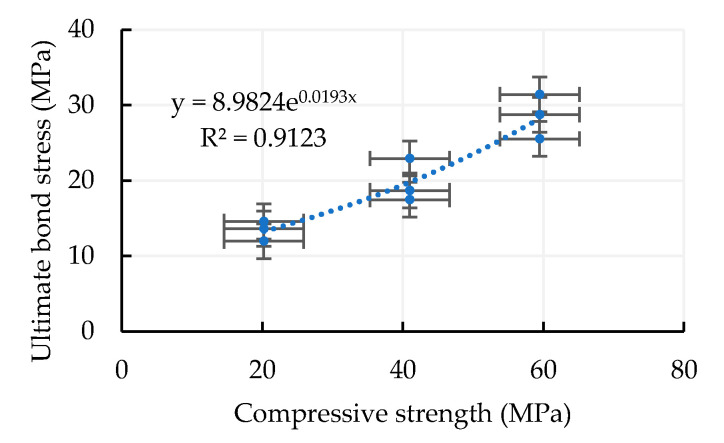
The proposed univariate regression prediction model of bond strength.

**Figure 12 materials-13-03701-f012:**
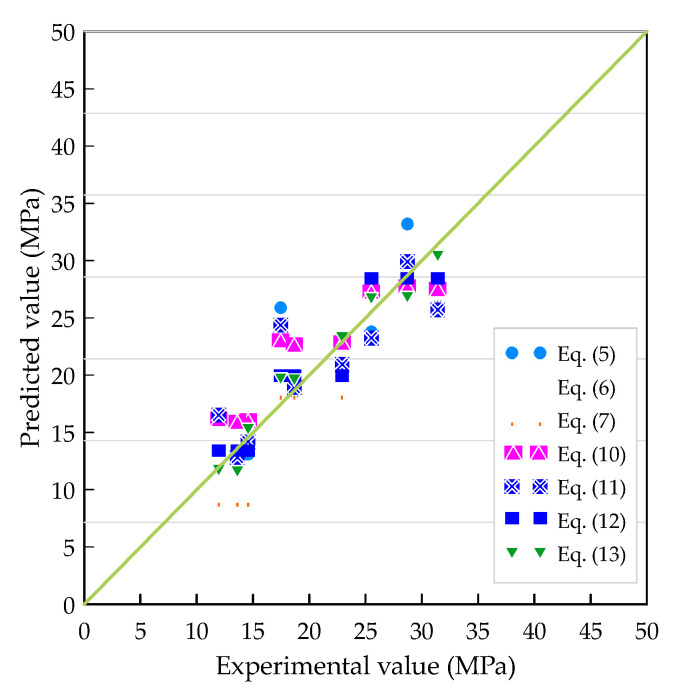
Comparison of prediction models of ultimate bond strength.

**Figure 13 materials-13-03701-f013:**
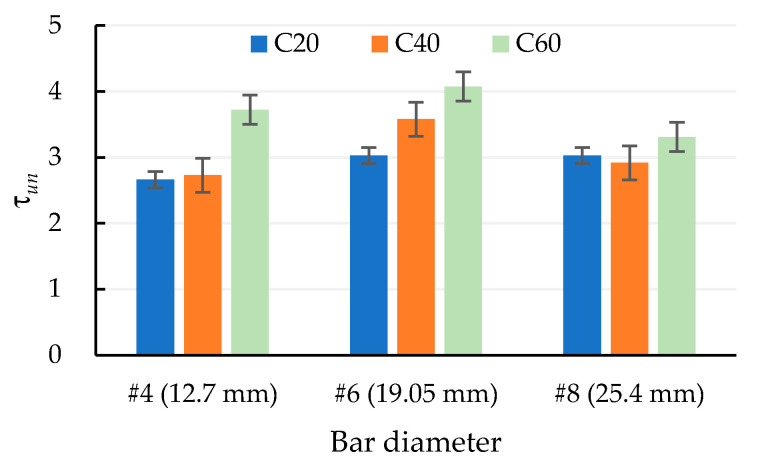
Normalized ultimate bond strength of different rebar diameters and concrete strength grades.

**Figure 14 materials-13-03701-f014:**
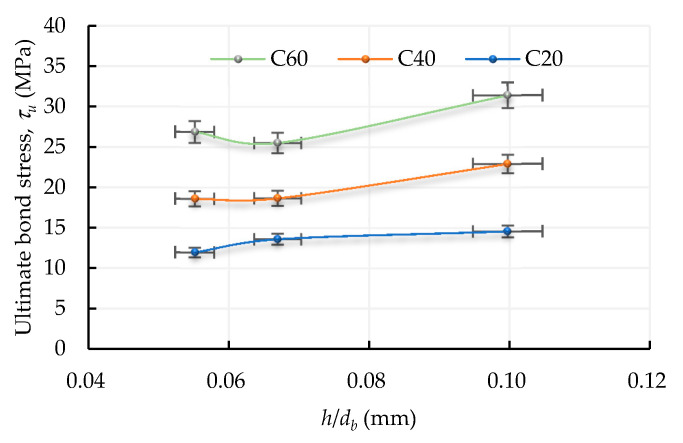
Relationship between $h/d_{b}$ and $\tau_{u}$.

**Figure 15 materials-13-03701-f015:**
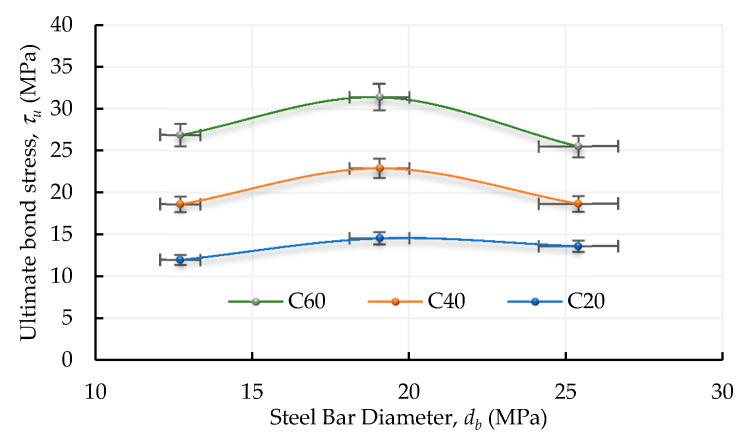
Relationship between the rebar diameter and ultimate bond stress for various concrete strengths.

**Figure 16 materials-13-03701-f016:**
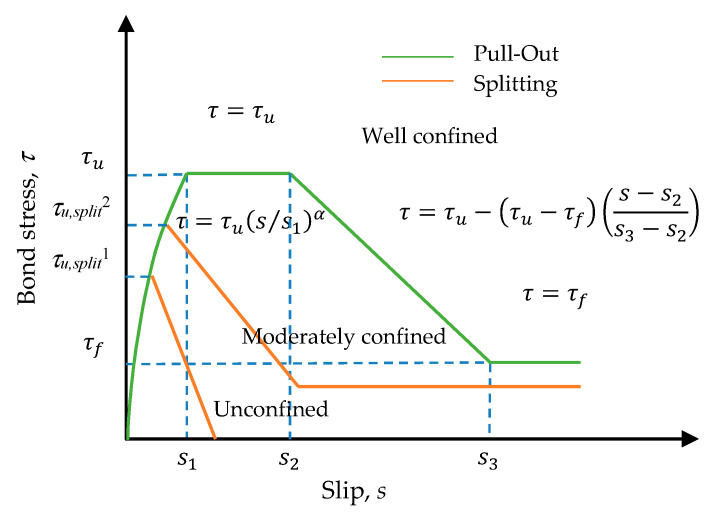
Analytical bond stress–slip relationship (CEB-FIP Model Code 2010).

**Figure 17 materials-13-03701-f017:**
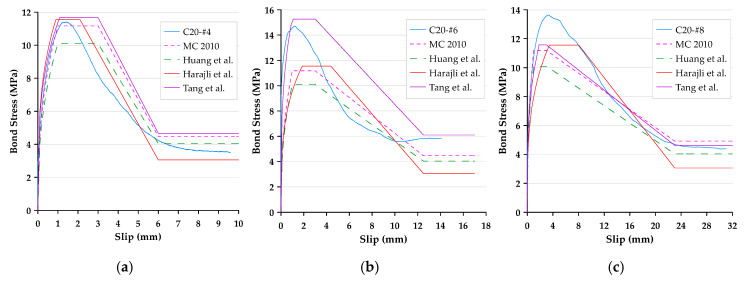
Comparison of the local bond stress–slip curve produced with different prediction models: (**a**) C20-#4, (**b**) C20-#6, and (**c**) C20-#8 (for specimens with low-strength concrete).

**Figure 18 materials-13-03701-f018:**
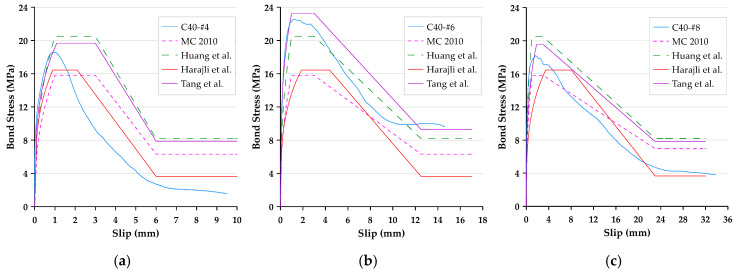
Comparison of the local bond stress–slip curve produced with different prediction models: (**a**) C40-#4, (**b**) C40-#6, and (**c**) C40-#8 (for specimens with medium-strength concrete).

**Figure 19 materials-13-03701-f019:**
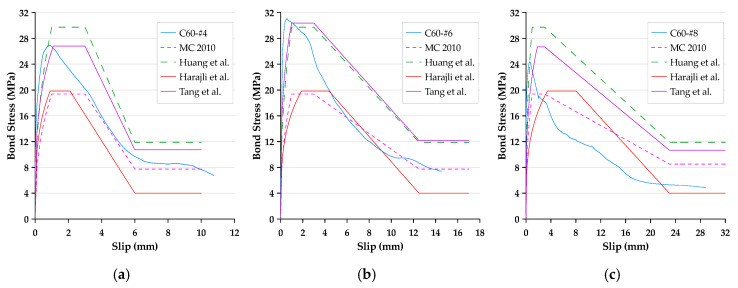
Comparison of the local bond stress–slip curve produced with different prediction models: (**a**) C60-#4, (**b**) C60-#6, and (**c**) C60-#8 (for specimens with high-strength concrete).

**Figure 20 materials-13-03701-f020:**
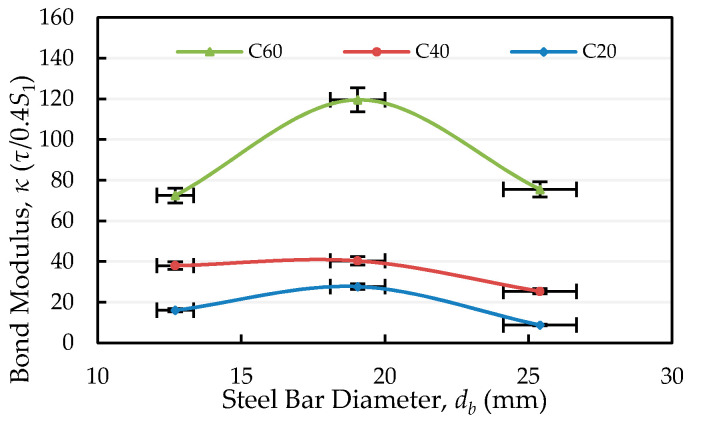
Comparison of the bond modulus under the same concrete compressive strength.

**Figure 21 materials-13-03701-f021:**
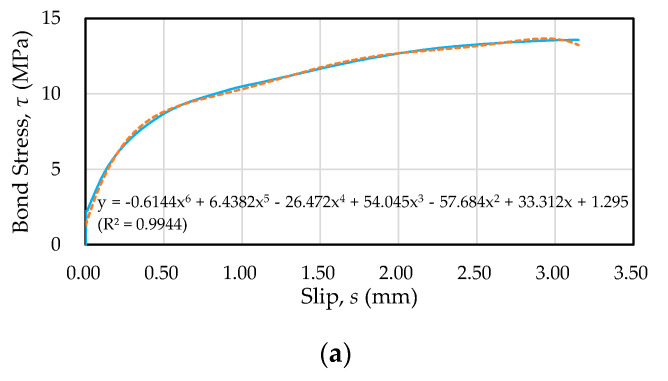
Comparison of **the** local bond stress–slip curve versus fitting curve for specimens with the #8 rebar: (**a**) C20, (**b**) C40, and (**c**) C60.

**Table 1 materials-13-03701-t001:** Physical properties of coarse/fine aggregate.

Aggregate Type	Specific Weight (SSD)	Water Absorption (SSD) (%)	Unit Weight (Dry-Rodded) (kg/m^3^)	FM
Coarse aggregate	2.63	1.24	1532	-
Fine aggregate	2.56	1.33	-	2.75

Notes: SSD, saturated surface-dry condition; FM, fineness modulus.

**Table 2 materials-13-03701-t002:** Basic properties of superplasticizer.

Type	Specific Weight	pH Value	Solid Composition (%)
HPC 1000	1.20	7 ± 1	3.37
MTP A40	1.13	7 ± 1	-

**Table 3 materials-13-03701-t003:** Basic properties of rebars.

Bar No.	Nominal Dia. (mm)	Nominal Cross Section Area (cm^2^)	Rib Distance (mm)	Rib Width (mm)	Rib Height (mm)	Elastic Modulus (GPa)
4	12.70	1.27	8.3	2.1	0.7	204
6	19.05	2.85	12.1	3.7	1.9	207
8	25.40	5.07	30.4	3.7	1.7	205

**Table 4 materials-13-03701-t004:** Concrete mix design composition.

Mix No.	Water/Cement Ratio (W/C)	Cement (kg/m^3^)	Water (kg/m^3^)	Aggregate(kg/m^3^)		SP (kg/m^3^)	Dry Unit Weight (kg/m^3^)
				FA	CA		
C20	0.76	267	203	772	1054	0	2147
C40	0.52	390	203	670	1054	0.78	2194
C60	0.32	591	189	523	1063	6.50	2301

Note: FA, fine aggregate; CA, coarse aggregate; SP, superplasticizer (HICON HPC 1000 for C40 and HICON MTP A40 for C60).

**Table 5 materials-13-03701-t005:** Mechanical properties of concrete.

Mix No.	Slump (cm)	Compressive Strength (MPa)	Splitting Strength (MPa)	Elastic Modulus (GPa)
C20	17	20.20	2.40	23.32
C40	16	40.97	2.91	30.22
C60	17	59.46	3.23	30.72

**Table 6 materials-13-03701-t006:** Test result of ultimate bond stress.

Specimen No.	$\mathrm{Ultimate}\;\mathrm{Bond}\;\mathrm{Stress}\;\tau_{u}\;(\mathrm{MPa})$		
	Specimen1	Specimen2	Average
C20-#4	12.52	11.42	11.96
C20-#6	14.71	14.42	14.57
C20-#8	13.58	13.63	13.61
C40-#4	16.30	18.61	17.46
C40-#6	23.26	22.57	22.92
C40-#8	19.14	18.20	18.67
C60-#4	30.55	26.88	28.72
C60-#6	31.04	31.79	31.41
C60-#8	24.35	26.68	25.52

**Table 7 materials-13-03701-t007:** Comparison of ultimate bond stress between the test result and prediction models.

Specimen No.	Test Results (MPa)	Results of Prediction Models (MPa)						
		Equation (5)	Equation (6)	Equation (7)	Equation (10)	Equation (11)	Equation (12)	Equation (13)
C20-#4	11.96	11.24 (−6.05)	9.09 (−24.00)	16.16 (35.12)	16.15 (35.04)	16.52 (38.10)	13.26 (10.91)	11.68 (−2.32)
C20-#6	14.57	11.24 (−22.88)	9.09 (−37.61)	13.10 (−10.06)	16.02 (9.96)	14.19 (−2.58)	13.26 (−8.96)	15.27 (4.80)
C20-#8	13.61	11.24 (−17.44)	9.09 (−33.21)	11.58 (−14.95)	15.89 (16.75)	12.83 (−5.72)	13.26 (−2.54)	11.57 (−15.00)
C40-#4	17.46	16.00 (−8.35)	18.44 (5.59)	25.89 (48.30)	23.00 (31.74)	24.37 (39.57)	19.81 (13.44)	19.67 (12.67)
C40-#6	22.92	16.00 (−30.18)	18.44 (−19.56)	21.00 (−8.39)	22.82 (−0.45)	20.94 (−8.63)	19.81 (−13.59)	23.26 (1.48)
C40-#8	18.67	16.00 (−14.29)	18.44 (−1.25)	18.55 (−0.65)	22.63 (21.21)	18.93 (1.40)	19.81 (6.08)	19.56 (4.76)
C60-#4	28.72	19.28 (−32.88)	26.76 (−6.83)	33.19 (15.57)	27.71 (−3.52)	29.91 (4.14)	28.30 (−1.46)	26.79 (−6.74)
C60-#6	31.41	19.28 (−38.63)	26.76 (−14.81)	26.91 (−14.31)	27.49 (−12.49)	25.70 (−18.17)	28.30 (−9.90)	30.37 (−3.30)
C60-#8	25.52	19.28 (−24.46)	26.76 (4.85)	23.78 (−6.83)	27.26 (6.83)	23.24 (−8.95)	28.30 (10.89)	26.67 (4.52)

Note: The value in brackets is the percentage error.

**Table 8 materials-13-03701-t008:** Parameter values of the prediction model for the bond stress–slip relationship.

Parameter	CEB-FIP Model [15]	Huang et al. [37]	Harajli et al. [40]	This Study
	Confined Concrete	Normal Strength Concrete	Concrete	Confined Concrete
$s_{1}$	1.0 mm	1.0 mm	0.15 Distance bet. ribs	Equation (15)
$s_{2}$	3.0 mm	3.0 mm	0.35 Distance bet. ribs	3.0 mm
$s_{3}$	Distance bet. ribs	Distance bet. ribs	Distance bet. ribs	Distance bet. ribs
*α*	0.4	0.4	0.3	0.3
$\tau_{u}$	$2.5\sqrt{f_{c}^{\prime }}$	$0.4f_{cm}$	$2.57\sqrt{f_{c}^{\prime }}$	Equation (13)
$\tau_{f}$	0.4$\tau_{u}$	0.4$\tau_{u}$	$0.9\sqrt{f_{c}^{\prime }}$	0.4$\tau_{u}$

**Table 9 materials-13-03701-t009:** Bond modulus for specimens with various bar diameters.

Mix No.	Bond Modulus, κ (τ/0.4*s*_1_)		
	#4 bar	#6 bar	#8 bar
C20	18.9	24.7	8.8
C40	35.5	40.3	25.3
C60	72.5	119.6	53.5

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
