# Peer review of "Modeling Local Bond Stress–Slip Relationships of Reinforcing Bars Embedded in Concrete with Different Strengths"

_materials, 2020, doi:10.3390/ma13173701_

Round 1
Reviewer 1 Report
The manuscript presents useful results and us generally well written. Technical issues I think you need to address are:
1) Adequacy of your number of specimens: Commonly investigators would replicate each type of specimen more than twice, with six replicates being a statistically acceptable number for estimation of an average response. Basically I’m asking you to justify/prove what you did is adequate.
2) Comparison with other predictions: Model codes give guidance on calculation of boned strength and other properties of materials that account for variability of those properties. Therefore, do you really think it is fair for you to directly compare Your test results with CEB-FIP Model Code values?
Editorially speaking, although your manuscript is perfectly readable you tend to use long sentences which can become convoluted. I suggest you edit your manuscript and break very long sentences into two or more shorter sentences.
Author Response
Response to Reviewer 1 Comments
The manuscript presents useful results and us generally well written. Technical issues I think you need to address are:
Point 1: Adequacy of your number of specimens: Commonly investigators would replicate each type of specimen more than twice, with six replicates being a statistically acceptable number for estimation of an average response. Basically, I’m asking you to justify/prove what you did is adequate.
Response: Thanks for the reviewer’s reminders and suggestions. In the experimental work of this article, the compressive strength, split tensile strength and elastic modulus of each concrete mixture were the average of three samples. As for the pull-out experiment, Table 6 presents the experimental results of ultimate bond stress for each combination of concrete strength and rebar diameter, which was the average of two nominally identical test specimens.
Point 2: Comparison with other predictions: Model codes give guidance on calculation of boned strength and other properties of materials that account for variability of those properties. Therefore, do you really think it is fair for you to directly compare your test results with CEB-FIP Model Code values?
Response: The general concrete design code aims to compulsorily standardize the design of concrete structures in engineering construction to ensure the standardization and safety of various types of engineering construction. In order to avoid complicated structure calculation process, most specifications adopt simplified design methods. But for the sake of safety, most of the variables will be considered conservatively in the simplification process, making the structural design unable to meet the economic requirements. However, through the research of scholars and experts, most of the regulations are constantly revised to meet the actual situation. Therefore, many studies have adopted this method, and this article is no exception.
Point 3: Editorially speaking, although your manuscript is perfectly readable you tend to use long sentences which can become convoluted. I suggest you edit your manuscript and break very long sentences into two or more shorter sentences.
Response: Thanks to the reviewer’s valuable suggestions, the long sentence has been divided into two or more shorter sentences in the revised version of the manuscript.

Reviewer 2 Report
1) Many authors are writing about the concrete production, however, most of your citations are from Asia. When I am browsing last e.g. “Materials” manuscripts the topic of concrete and waste for concrete are often discussed. To increase the rank of an article, it may be worth quoting authors from around the world f.e.:
- doi:10.3390/ma13143189
- doi:10.3390/ma13081827
This is only suggestion, but I think it will significantly increase the scientific level of the article.
2) On figures error bars must be added, because your values are very similar.
3) What about the test on fresh mortar?
4) Please explain the method of concrete recipe calculation.
5) Why You coos concrete with compressive strength 20, 40 and 60 MPa?
6) Please add more information about superplasticizers, e.g. what kind of substance you use, and and in what proportion/ratio? Additionally, why you use supreplasticizers?
7) You write that you use local aggregate? Please explain what kind of aggregate? Maybe you have done sieve curve?
8) You should expand the paragraph relating to the preparation of the concrete mixture about the time, vibration time, rotation, type of mixing,
9) Why did you not determine the flexural properties of concrete ?
10) Please explain what is mean "local type I portland cement". What is the characteristic compressive strength of this cement?
11) According to which standards the tests were performed?
Author Response
Response to Reviewer 2 Comments
Point 1: Many authors are writing about the concrete production, however, most of your citations are from Asia. When I am browsing last e.g. “Materials” manuscripts the topic of concrete and waste for concrete are often discussed. To increase the rank of an article, it may be worth quoting authors from around the world f.e.:
doi:10.3390/ma13143189
doi:10.3390/ma13081827
This is only suggestion, but I think it will significantly increase the scientific level of the article.
Response: Thanks to the reviewer’s valuable suggestions, articles published in Europe and the United States have been added to the references. Among the revised manuscripts, 35 of the 53 documents were published in Europe and the United States.
Point 2: On figures error bars must be added, because your values are very similar.
Response: Error bars have been added to the figures in the revised version of the manuscript.
Point 3: What about the test on fresh mortar?
Response: This article mainly focuses on concrete materials to discuss local bond stress-slip relationships of reinforcing bars embedded in concrete with different strengths, and does not make mortar samples. The slump values of each concrete mixture are listed in Table 5. It can be seen from Table 5 that the slump of the three groups of concrete proportions is approximately the same, about 16-17 cm.
Point 4: Please explain the method of concrete recipe calculation.
Response: The proportion of concrete must be selected to provide the necessary placeability, density, strength, and durability for the specific application. This article referred to the ACI 211.1-91 specification for concrete mix design, and adjusted the composition after trial mixing.
Point 5: Why you coo concrete with compressive strength 20, 40 and 60 MPa?
Response: The literature shows that the bond strength is closely related to the compressive strength of concrete. In view of the fact that some predictive formulas are too conservative or overestimated for the prediction of the bond strength of concrete with different strengths, three concretes with different compressive strengths (representing low, medium and high grades respectively) are planned for further discussion and analysis.
Point 6: Please add more information about superplasticizers, e.g. what kind of substance you use, and in what proportion/ratio? Additionally, why you use supreplasticizers?
Response: To improve the workability of concrete, two superplasticizers were used. HICON HPC 1000 complies with the American ASTM C494 Type D regulations, with a PH value of 7±1, a specific gravity of 1.2, and a solid content of 3.37%. HICON MTP A40 complies with the American ASTM C494 Type G regulations, with a PH value of 7±1 and a specific gravity of 1.13. The dosage of HICON HPC 1000 was 0.2% of cement. The dosage of HICON MTP A40 was 1.1% of cement.
Point 7: You write that you use local aggregate? Please explain what kind of aggregate? Maybe you have done sieve curve?
Response: The aggregate used in this article was a siliceous aggregate. The coarse aggregate was crushed stone with a maximum particle size of 19 mm and fine aggregate was natural river sand. Their physical properties are listed in Table 1.
Point 8: You should expand the paragraph relating to the preparation of the concrete mixture about the time, vibration time, rotation, type of mixing,
Response: In the revised version of the manuscript, it has been explained as follows:
“During mixing, the cement, fine aggregates, and coarse aggregates were first blended in the biaxial mixer at a rate of 45 revolutions per minute for about 1 minute. Then the water and superplasticizer were added in the mixer and blended for about 1.5 minutes.”
Point 9: Why did you not determine the flexural properties of concrete?
Response: The subject of this article is the bond strength, which is less related to the flexural strength of concrete. On the contrary, the splitting strength is more influential, so the splitting strength test was carried out. The results are shown in Table 5.
Point 10: Please explain what is mean "local Type I Portland cement". What is the characteristic compressive strength of this cement?
Response: In Taiwan, there are 5 types of cement. The first type is Type I Portland cement (ordinary cement), which is suitable for general use. The 28-day characteristic compressive strength of Type I Portland cement is 42.4 MPa.
Point 11: According to which standards the tests were performed?
Response: In the revised version of the manuscript, it has been explained as follows:
“According to ASTM C143 [50], ASTM C39 [51], ASTM C496 [52] and ASTM C469 [53], the slump, compressive strength, splitting strength and elastic modulus of concrete were tested respectively.”

Reviewer 3 Report
Dear authors, thank you for the interesting paper.
My comments are:
- line 69: "Eligehausen et al. [8,9] ... - only reference [8] is Eligehausen et al., the reference [9] is not Eligehausen et al. but Oh & Kim,
- formulas (12,13) - are the empirical formulas or the coefficients has units?
- figures 14 and 15 - should it be lines with dots? or dots are other values as lines? Difficult to tell if they are the same numbers or not, why are the dots different from the lines?
- formula (15) - is it also empirical formula? Units do not sit. Slip "s" is in millimeters, on right side is s3xs3 [mm2].

Author Response
Response to Reviewer 3 Comments
Dear authors, thank you for the interesting paper.
My comments are:
Point 1: Line 69: "Eligehausen et al. [8,9] ... - only reference [8] is Eligehausen et al., the reference [9] is not Eligehausen et al. but Oh & Kim,
Response: Thanks for the reviewer’s correction, the revised version of the manuscript has been changed as follows:
“Eligehausen et al. [8] set a shorter anchor length ( , where is the anchor length and is the diameter of the reinforcing bar) in their experimental research. Its purpose is to reduce the influence caused by an uneven bond stress distribution along the embedded length of the steel bar. Therefore, that the measured test results could be close to the true local bond stress. Oh and Kim [9] proposed a realistic model of the bond stress-slip relationship under repeated loads. The test showed that if bond failure did not occur, the bond strength and the slip at peak bond stress were not influenced much by repeated loading. However, the values of loaded slip and residual slip increased with increasing load cycles.”
Point 2: Formulas (12,13) - are the empirical formulas or the coefficients has units?
MPa
Response: In Formulas (12,13), the unit of ultimate bond stress is MPa, and its determination coefficient has no unit. The revised version of the manuscript has been changed as follows:
(MPa), |
(12) |
(MPa), |
(13) |
where compressive strength of concrete, c is the thickness of concrete cover, and is the diameter of the rebar. It can be seen from Figure 11 that the proposed univariate regression prediction model of bond strength has a good correlation coefficient. The coefficient of determination (R2) of Equation (13) is 0.9568.
Point 3: Figures 14 and 15 - should it be lines with dots? or dots are other values as lines? Difficult to tell if they are the same numbers or not, why are the dots different from the lines?
Response: In Figures 14 and 15, the dot represents the actual test result, and the line represents the change trend of ultimate bond stress with the variable.
Point 4: Formula (15) - is it also empirical formula? Units do not sit. Slip "s" is in millimeters, on right side is s3xs3 [mm2].
Response: Statistical regression analysis can establish the relationship between the dependent variable and the independent variable, expressed by a mathematical model. In the regression equation, the scale of the dependent variable and the independent variable can be different. In the original data, if the scale is inconsistent, use standardized coefficients, that is, standardize the original independent variables. The standardized variables will not be affected by different scales. The regression coefficient calculated from the standardized independent variable is called the β coefficient. Whether the regression model is effective and its explanatory power is generally based on the coefficient of determination (R2) as an indicator. The regression equations in this article used standardized coefficients.

Round 2
Reviewer 1 Report
You have addressed my previous comment adequately.
Reviewer 2 Report
All my suggestions were added and included in the text. In this form I accept the article for printing process.